# Revisiting density-dependent fecundity in schistosomes using sibship reconstruction

**M. Inês Neves**[1,2]*, **Charlotte M. Gower**[2], **Joanne P. Webster**[1,2], **Martin Walker**[1,2]

**1** Department of Pathobiology and Population Sciences, Royal Veterinary College, University of London, United Kingdom, **2** London Centre for Neglected Tropical Disease Research, Imperial College London Faculty of Medicine, London, United Kingdom

* mneves@rvc.ac.uk

**Data Availability Statement:** The Tanzanian data are fully available in accompanying S1 Data file. The data associated with the Zanzibar project are

## Abstract

The stability of parasite populations is regulated by density-dependent processes occurring at different stages of their life cycle. In dioecious helminth infections, density-dependent fecundity is one such regulatory process that describes the reduction in egg production by female worms in high worm burden within-host environments. In human schistosomiasis, the operation of density-dependent fecundity is equivocal and investigation is hampered by the inaccessibility of adult worms that are located intravascularly. Current understanding is almost exclusively limited to data collected from two human autopsy studies conducted over 40 years ago, with subsequent analyses having reached conflicting conclusions. Whether egg production is regulated in a density-dependent manner is key to predicting the effectiveness of interventions targeting the elimination of schistosomiasis and to the interpretation of parasitological data collected during monitoring and evaluation activities. Here, we revisit density-dependent fecundity in the two most globally important human *Schistosoma* spp. using a statistical modelling approach that combines molecular inference on the number of parents/adult worms in individual human hosts with parasitological egg count data from mainland Tanzania and Zanzibar. We find a non-proportional relationship between *S. haematobium* egg counts and inferred numbers of female worms, providing the first clear evidence of density-dependent fecundity in this schistosome species. We do not find robust evidence for density-dependent fecundity in *S. mansoni* because of high sensitivity to some modelling assumptions and the lower statistical power of the available data. We discuss the strengths and limitations of our model-based analytical approach and its potential for improving our understanding of density dependence in schistosomiasis and other human helminthiases earmarked for elimination.

## Author summary

Schistosomiasis is a devastating disease of poverty currently estimated to infect over 220 million people. It is caused by parasitic worms (blood flukes) that live for, on average, 5–7 years inside the blood vessels of infected hosts and produce hundreds of eggs daily. Whether egg production is regulated in a density-dependent manner, and if so under

openly available in ClinEpiDB: "Study: SCORE Zanzibar S. haematobium Cluster Randomized Trial" and can be found at https://clinepidb.org/ce/app/record/dataset/DS_eddb4757ba.

**Funding:** This article is to be published and paid by Chronos under the Gates Foundation Open Access Policy under OPP 50816 Schistosomiasis Consortium for Operational Research and Evaluation (SCORE). MIN acknowledges funding from a Royal Veterinary College, University of London PhD Studentship. CMG acknowledges funding from a Royal Society Dorothy Hodgkin Clore Fellowship. Collection and primary analyses of the mainland Tanzania dataset was funded by The Royal Society and The Bill and Melinda Gates Foundation (BMGF) via the Schistosomiasis Control Initiative (PI: JPW) and of the Zanzibar dataset, by the BMGF via University of Georgia Research Foundation, Inc (UGARF) for the Schistosomiasis Consortium for Operational Research and Evaluation (SCORE) project (Population Genetics Grant Refs RR374-053/5054146 & RR374-053/4785426 PI: JPW). The funders had no role in study design, data collection and analysis, decision to publish, or preparation of the manuscript.

**Competing interests:** The authors have declared that no competing interests exist.

what conditions, has been controversial for schistosomiasis, and investigation is hampered due to the inaccessible location of adult worms. Resolving this fundamental question is important because density dependencies determine the resilience of helminthiases to interventions. Here, we have revisited this longstanding and unresolved question of density-dependent fecundity in human schistosomes using a novel statistical modelling approach that combines information from molecular and parasitological data. We report the first clear evidence of density-dependent fecundity in *S. haematobium*, the causative agent of millions of cases of urogenital schistosomiasis. Our findings are of critical importance both to mathematical modellers predicting the impact of interventions and to public health policy makers striving to meet the 2030 elimination targets for schistosomiasis. This study also serves to illustrate a new biostatistical approach that could be applied to investigate density dependencies in other helminthiases where adult parasites are inaccessible.

## Background

Schistosomiasis is a devastating neglected tropical disease (NTD) caused by trematode parasites, currently estimated to infect at least 220 million people, 90% of whom live in sub-Saharan Africa (SSA) [1, 2]. *Schistosoma mansoni* and *S. haematobium* are the main species causing intestinal and urogenital schistosomiasis in humans. Transmission occurs through contact with freshwater habitats of the intermediate snail hosts (*Biomphalaria* and *Bulinus* spp. respectively) which have been contaminated by eggs released in faeces or urine. Chronic schistosome infections can cause significant morbidity characterised by a broad range of pathologies including, but not exclusive to, anaemia, chronic pain, stunting, cystitis, genital lesions, irreversible organ damage and cancer [2–5]. Globally, schistosomiasis is the second most important parasitic disease, after malaria, in terms of socioeconomic impact [6].

Preventive chemotherapy by mass drug administration (MDA) with praziquantel is the World Health Organization's (WHO) recommended strategy for controlling schistosomiasis and has been implemented across much of SSA since 2002 [7, 8]. The initial success of MDA led the WHO to set ambitious goals for the control of schistosomiasis by 2020 [9], and its elimination as a public health problem in all endemic countries by 2030 [10]. Whilst the feasibility of reaching these goals will be heterogeneous among schistosomiasis foci and will be particularly challenging where transmission is intense [11], elucidating the basic biology and fitness strategies available to these parasites will be critical in terms of predicting and evaluating MDA impact and, if necessary, modifying strategies accordingly.

Mathematical modelling is increasingly being used to inform high-level decision making on intervention strategies against schistosomiasis, and NTDs more generally [12, 13]. Yet the utility of models hinges fundamentally on our understanding of the underlying population biology of the pathogen. In schistosomiasis, a key unresolved and longstanding question is whether and to what extent density dependence reduces egg production by female worms in high worm burden within-host environments [14–17]. Density-dependent fecundity regulates the size of parasite populations and enhances their resilience to intervention [18–21]. Hence, elucidating the worm-egg relationship and the existence or not of density dependence in schistosomiasis is of utmost importance to both modelling the effectiveness of intervention strategies and interpreting the routine egg count data collected for monitoring and evaluation purposes.

The challenge to identifying density-dependent fecundity in schistosomiasis stems from the inaccessible intravascular location of the adult worms. Unlike the majority of intestinal helminths, which can be expelled chemotherapeutically and counted (e.g. [22–27]), schistosomes in humans can only be enumerated directly at autopsy. Published studies, conducted over 40 years ago, did attempt this [28, 29], although subsequent analyses of these worm-egg datasets have resulted in conflicting conclusions on the operation of density dependence [14–17]. A new alternative, indirect, approach involves the identification of parental genotypes by genetic analysis of (accessible) schistosome offspring (miracidia hatched from eggs in urine or faeces) [30–32]. This technique—a branch of parentage analysis called sibship reconstruction—permits quantification of the number of unique parental genotypes by dividing a sample of offspring genotypes into groups of full siblings (monogamous mating) or groups of full and half siblings (polygamous mating) [33–36].

Here we revisit density-dependence in schistosomes by analysing the functional relationship between egg counts and the number of female worms within a host, inferred by sibship reconstruction. We use paired egg count and genotypic data on *S. haematobium* collected in Zanzibar [37] and *S. mansoni* collected in mainland Tanzania [31] respectively to evaluate density-dependent fecundity in both species, making use of our recently developed statistical approach [38] to adjust for inherent bias and uncertainty in estimates of female worm burden. We discuss the strengths and limitations of our model-based approach, its potential for enhancing our understanding of density dependence, and the consequences of this fundamental population process on intervention design and interpretation of routine monitoring and evaluation data.

## Methods

### Epidemiological studies

We analysed egg count and miracidial genotypic data derived from two epidemiological studies: one conducted in Zanzibar as part of the 'Zanzibar Elimination of Schistosomiasis Transmission' (ZEST) alliance and the 'Schistosomiasis Consortium for Operational Research and Evaluation' (SCORE) [31, 39] and the other conducted in mainland Tanzania, as part of Schistosomiasis Control Initiative (SCI) activities [7, 31]. The SCORE/ZEST study was a cluster-randomised trial involving three study arms (with differing control pressures / intervention strategies) implemented in 90 randomly selected shehias (small administrative regions) on both islands of the Zanzibar archipelago, Pemba and Unguja [37]. Parasitological and genotypic data on *S. haematobium* were collected from 224 children and adults in 2012, before onset of the clinical trial, and 214 children and adults in 2016, after 10 rounds of biannual MDA. The SCI study was undertaken in the Lake Victoria region of Tanzania, collecting parasitological and genotypic data on *S. mansoni* from 151 schoolchildren attending two primary schools between 2005 and 2010, before and during MDA (delivered annually in one school with a missed treatment in 2008, and delivered in 2005, 2007 and 2010 in the other school) [31]. In both studies, samples were collected two months before treatment with praziquantel.

### Parasitological methods

In the Zanzibarian (SCORE/ZEST) study, urine samples were collected from each individual and urine filtration was used for egg identification and quantification of the intensity of *S. haematobium* infection. Infection intensity was expressed as eggs per 10 ml of urine. In the Tanzanian (SCI) study, duplicate Kato-Katz thick smears were prepared from individual stool samples from each child, eggs were counted, and infection intensity of *S. mansoni* was expressed as eggs per gram (epg) of stool. *S. mansoni* eggs were then purified from separately

prepared stool samples from each infected child and hatched into individual miracidia [40]. Egg hatching and isolation of *S. mansoni* and *S. haematobium* was performed by concentrating eggs from all infected urine samples by filtration using a Pitchford Funnel, rinsing and transferring into a clean Petri dish containing mineral water and exposing eggs to light to facilitate hatching of miracidia [41].

## Molecular methods

In both studies, the number of unique female *S. haematobium* and *S. mansoni* genotypes within each individual, *n*, was estimated by analysing multiplexed microsatellite genotypic data of individual miracidia (hatched from eggs) using sibship reconstruction methods [36]. Sibship reconstruction is a category of parentage analysis which can be used to estimate the number of parents when genetic data are available on offspring only [33–36]. Essentially, data on neutral genetic markers are used to divide offspring into groups of full siblings (monogamous mating) or groups of full siblings and half siblings (polygamous mating) to reconstruct and identify unique (male and/or female) parental genotypes. Hence the technique can be used to estimate worm burdens ([30], and see for examples [31, 32, 42]) after statistical adjustment for the number of offspring (here, miracidia) sampled [38].

The number of miracidia sampled per individual, *m*, ranged from 1 to 28 in the Zanzibarian (*S. haematobium*) study, and from 1 to 20 in the Tanzanian (*S. mansoni*) study. Complete details on the molecular analysis and sibship reconstruction can be found in studies by Gower et al. 2017 [31] and by the ZEST alliance and SCORE [31, 39].

## Statistical modelling

We fitted log-linear statistical models of the general form

$$\log(\mathbf{Y} + 1) = \beta_0 + \beta_1 \log(\mathbf{N}) + \gamma\mathbf{X} + \boldsymbol{e}, \tag{1}$$

where **Y** is a vector of observed eggs per unit volume of faeces or urine (i.e. epg or eggs per 10ml urine), **N** is a vector of inferred female worm burdens per host and the coefficients $\beta_0$ and $\beta_1$ denote, respectively, the per worm fecundity (i.e. the egg output for $N = 1$) and the direction and severity of density dependence (Fig 1). The error term $\boldsymbol{e}$ is assumed to be normally distributed, such that the of distribution of **Y** + 1 is log normal with *median* $\boldsymbol{\mu_Y} = \exp(\beta_0)\mathbf{N}^{\beta_1}\exp(\boldsymbol{\gamma}\mathbf{X})$. The matrix **X** comprises additional covariates, with corresponding coefficients denoted $\boldsymbol{\gamma}$. For the Zanzibarian study (*S. haematobium*), **X** comprised indicator variables for island (Pemba or Unguja), age group (child or adult), sex (male or female) and year (2012 or 2016). For the Tanzanian study (*S. mansoni*), **X** comprised indicator variables for school (Bukindo or Kisorya), sex (male or female) and year (2005, 2006 or 2010).

## Regression calibration

We used a regression calibration technique [43] to integrate uncertainty (measurement error) in *N* on the estimated coefficient of density dependence, $\beta_1$. We achieved this by simulating 1,000 datasets $\mathbf{N}_1, \mathbf{N}_2, \ldots, \mathbf{N}_{1000}$ from the posterior distribution defined in Neves et al. [38],

$$f(N|n, m) \propto f(n|N, m)f(N) \tag{2}$$

and refitting the regression model (Eq 1) to each replica dataset. The posterior $f(N|n, m)$ captures uncertainty in *N* derived from the number of unique female genotypes, *n*, being identified from a finite sample of miracidia, *m*. The most precise estimates of *N* are achieved when $n \ll m$ [38].

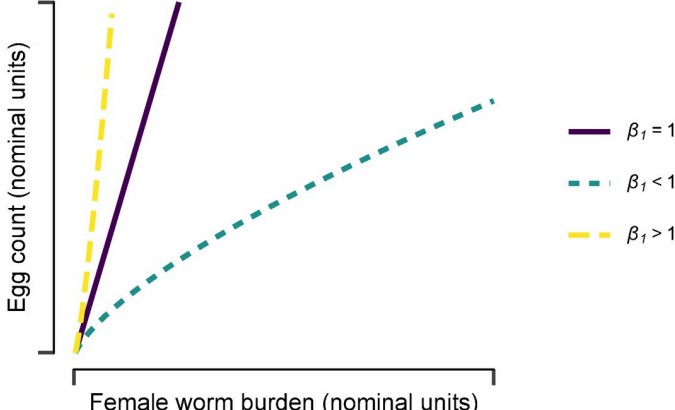

**Fig 1. Functional relationship between schistosome egg count and female worm burden inferred from sibship reconstruction.** The log-linear regression model implies a relationship between the median egg count per host, $\mu_Y$, and the female worm burden $N$ of the form $\mu_Y = \beta_0 N^{\beta_1}$, where $\beta_0$ denotes the per worm egg production (fecundity) in the absence of density dependence (i.e. when $N = 1$) and $\beta_1$ governs the direction (and severity) of density dependence. The solid purple line indicates a proportional (density-independent) relationship for $\beta_1 = 1$, the dashed yellow line a facilitating (positive) density-dependent relationship for $\beta_1 > 1$ and the dotted blue line a constraining (negative) density-dependent relationship when $\beta_1 < 1$.

## Sensitivity analysis

We assumed that the prior distribution $f(N)$ was uniform with support $N \in [n, N_{\max}]$. That is, we assumed that the minimum number of female worms in a host was given by the $n$ unique genotypes identified by sibship reconstruction (i.e. there was no false identification of unique genotypes) and the maximum by parameter $N_{\max}$, which is required to ensure the posterior has a finite upper bound when $m/n \to 1$ [38]. The only information on the maximum number of schistosomes a human can (plausibly) harbour comes from two autopsy studies that counted a maximum of 350 adult female *S. mansoni* and 250 female *S. haematobium* directly from 103 and 197 people respectively (excluding individuals with Symmer's fibrosis) [28, 29]. However, to explore the impact of this assumption on the coefficients of density dependence we repeated the statistical modelling and regression calibration approaches for values of $N_{\max}$ ranging from 100 to 2,000.

## Results

The log-linear regression model structure permitted flexibility to identify constraining (negative) density dependence (density-dependent coefficient, $\beta_1 < 1$) proportionality (i.e. a linear, density-independent relationship, $\beta_1 = 1$) and facilitating (positive) density dependence ($\beta_1 > 1$) between per host egg count and (female) worm burden, $N$ (Fig 1). We integrated uncertainty associated with estimates of $N$ by re-fitting the regression model to 1,000 datasets simulated from the posterior distribution of $N$, for each of a range of values for $N_{\max}$ (as a sensitivity analysis).

For *S. haematobium*, we found that point estimates of the density-dependent coefficient (from each simulated dataset) and their associated upper 95% confidence limits are consistently less than 1 for a range of values for $N_{\max}$, indicating density-dependent fecundity (Fig 2). This effect is further illustrated in Fig 3, which shows the observed and model-fitted relationship between *S. haematobium* egg counts and inferred female worm burden (integrating uncertainty associated with $N$).

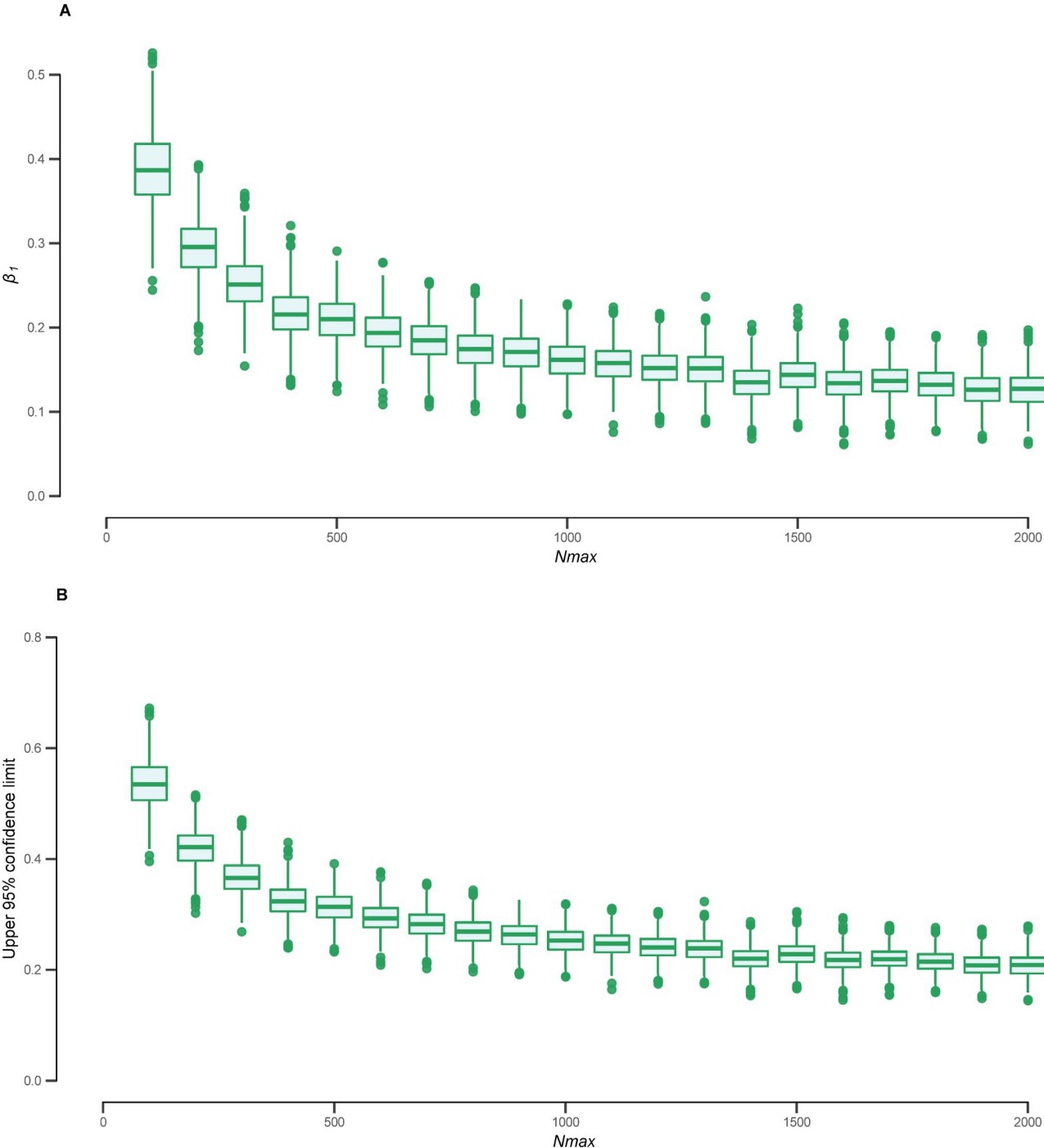

**Fig 2. Density-dependent coefficient estimated from *Schistosoma haematobium* egg count data and inferred numbers of female worms.** In panels A and B each point indicates, respectively, the point estimate, $\beta_1$, and upper 95% confidence limit of the coefficient of density dependence estimated by repeatedly fitting a log-linear regression model to 1,000 datasets. Each dataset was generated by sampling from the posterior distribution of the inferred number of female worms, $N$, assuming a different value $N_{max}$ for the upper bound of the uniform prior distribution of $N$. The box and whiskers depict the median, interquartile range and the 2.5th and 97.5th percentiles of the estimates. Values less than 1 indicate statistical support for density-dependent fecundity.

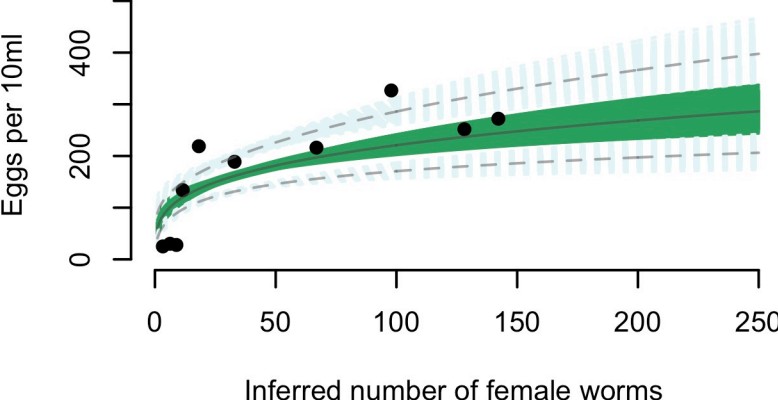

**Fig 3. The observed and fitted relationship between *Schistosoma haematobium* egg counts and inferred female worm burden.** The data points represent the expected value (mean) of the inferred number of female worms per host, $N$, binned by deciles. The green area indicates the range of model fits to each of 1,000 datasets simulated from the posterior distribution of $N$ assuming an upper bound of the uniform prior distribution of $N_{max} = 250$. The blue areas represent the range of 95% confidence intervals associated with each model fit. The solid grey line shows the mean of the model fits, and the dashed grey lines the mean of the upper and lower 95% confidence intervals. In this example fit, additional covariates are set to their reference value, i.e. for a child sampled in Pemba Island in 2012 and 2016.

We did not find consistent support for density-dependent fecundity for *S. mansoni*, with coefficient estimates varying substantially with different assumed values of $N_{max}$ (Fig A in S1 Text). For values of $N_{max} \gtrsim 500$, density dependence was indicated, but for values less than 500, uncertainty in our estimate of the density-dependent coefficient was too large to reach a definitive conclusion. The sensitivity of these results to $N_{max}$ is caused by the generally higher values of $m/n$ in the *S. mansoni* data (Fig 4).

A comprehensive list of coefficient estimates for each fitted model (summary statistics of coefficients estimated from the 1,000 datasets simulated for each value of $N_{max}$) can be found in Tables A and B in S1 Text. We also tested models that included interaction terms, permitting the severity of density dependence to vary with the other measured covariates. These models did not provide a sufficiently improved fit to the data to warrant being preferred over the simpler more parsimonious additive models, as indicated by likelihood-ratio tests (see Text A and Table C in S1 Text).

## Discussion

We revisited the longstanding and unresolved question of density-dependent fecundity in human schistosomes using an approach that combines traditional parasitological (egg count) data with data from contemporary genetic analyses that permit inference on female worm burden. We found, for the first time, clear evidence supporting density-dependent fecundity in *S. haematobium*, the cause of urogenital schistosomiasis. In *S. mansoni*, the cause of intestinal schistosomiasis, the analysis was more sensitive to the modelling assumptions, prohibiting a definitive conclusion on the operation of density-dependent fecundity. While our modelling approach has limitations, it provides a unique means of evaluating density dependence in schistosomes and can be extended to other human helminthiases where adult parasites are inaccessible. Density dependence is of fundamental importance to the population and transmission dynamics of schistosomiasis and other helminthiases, particularly in the context of global elimination efforts.

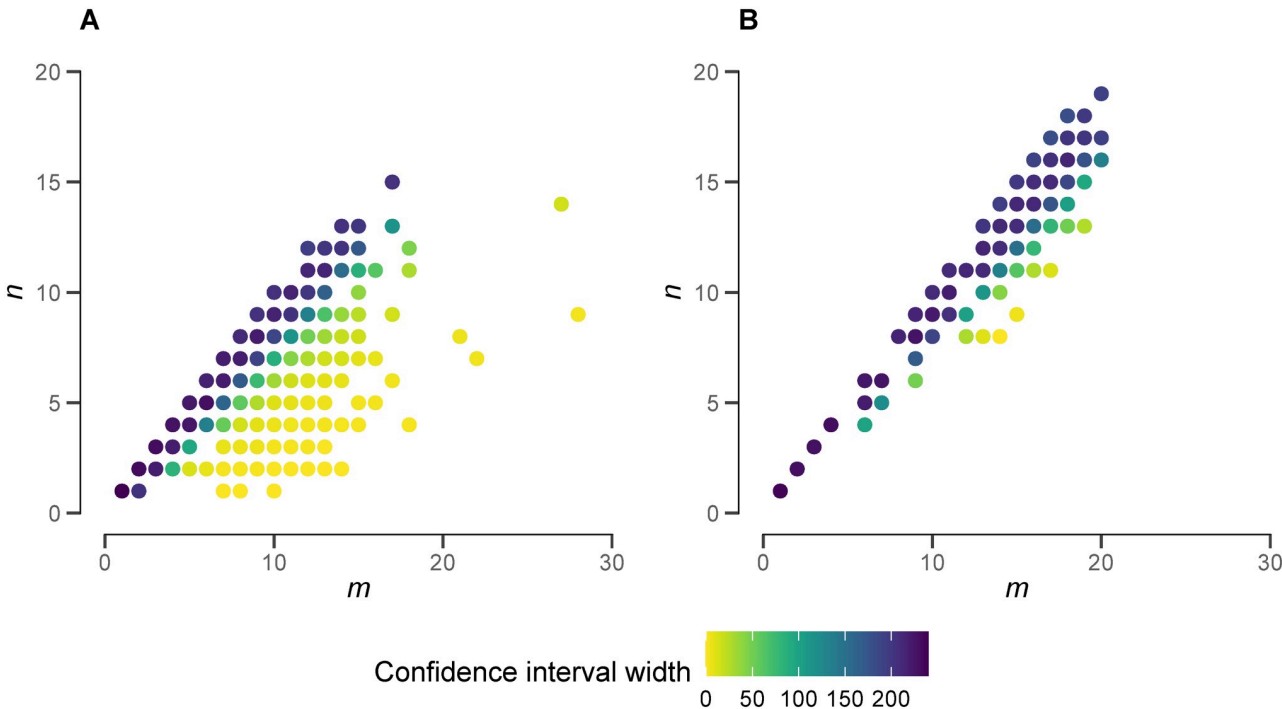

**Fig 4. Precision of the estimated female worm burden from $n$ unique genotypes identified from $m$ miracidia.** Points are plotted at the value of $m$ and $n$ from the *Schistosoma haematobium* [37] and *S. mansoni* [31] data in panels A and B respectively. Points are coloured according to the width of the 95% confidence interval associated with the posterior estimate of the female worm burden, $N$. The prior distribution for $N$ was assumed to be uniform with support $N \in [n, N_{max}]$, where $N_{max}$ was set to 250 for *S. haematobium* and 350 for *S. mansoni* on the basis of the maximum number of worms counted in a host from the Cheever et al. autopsy studies [28, 29]. Note in panel B, for *S. mansoni* a larger proportion of the data fall on or near the line of equality, $m/n = 1$.

## Comparison with previous findings

Density-dependent fecundity in schistosome infections has long been a controversial and debated topic. The phenomenon was initially reported in 1985 following analysis [16] of *S. mansoni* human autopsy data [29], but later refuted after new analyses of the same and additional autopsy data [14, 15, 17, 28]. In 2001, taking a different approach, Polman et al. [44] found that the relationship between circulating *S. mansoni* antigens and egg counts was not proportional, potentially indicative of density-dependent fecundity. However, the interpretation of this relationship relies on the assumption that measured antigen levels relate to worm burdens in a linear or proportional manner, which remains unclear.

A recent study by Gower et al. [31], using the same Tanzanian *S. mansoni* data as here, found a higher mean egg output per adult worm pair (inferred by sibship reconstruction) following 5 years of MDA treatment, which could be consistent with a relaxation of density-dependent constraints. In our study, we took a more conservative approach by accounting explicitly for the uncertainty in the estimated female worm burden [38], which is driven principally by the number of miracidia genotyped per host and the nominal maximum possible number of worms a host can harbour, $N_{max}$ (i.e. the upper bound of our prior assumption on $N$). Hence, while for values of $N_{max} \gtrsim 500$, we found evidence for density-dependent fecundity (Fig A in S1 Text)—consistent with the observations of Gower et al. [31]—we cannot reach a definitive conclusion because for $N_{max} \lesssim 500$ the upper uncertainty interval of the density-dependent coefficient frequently intersected 1.

A previous analysis of density-dependent fecundity in *S. haematobium* [28] (analysing 197 infected cases at autopsy) did not find significant evidence of density dependence. The apparently contradictory nature of our findings compared to the autopsy study may be explained partly by the different methodological approaches, and partly by differences in the representativeness of sampling. In the autopsy study, technical challenges of perfusion could lead to incomplete recovery of all worms in human cadavers. It is also possible that post-mortem extraction of urine leads to differential recovery of schistosome eggs than would be found when naturally expelled. The nature and immune status of individuals from the autopsy study is also very different from our study, since it was performed using deceased patients who had been terminally ill and had heavy worm burdens, whereas our study was conducted primarily using data from children attending primary schools. Our method of exploring density dependence is more indirect (but practicable), and necessarily invokes modelling assumptions (to which the results can be sensitive) which are fundamentally challenging to validate and which we fully acknowledge as a limitation. On the other hand, a major advantage of our approach is that it permits much more representative sampling of a host population compared to sampling at autopsy. This will be particularly important if factors related to the general health of the host (such as immunocompetence [45]) play a role in mediating density dependence.

## Limitations, uncertainty and validation

The sensitivity of our results to $N_{max}$ for the *S. mansoni* data—which we do not see for *S. haematobium*—stems from the difference in the ratio of the number of miracidia sampled to the number of unique female genotypes identified, $m/n$ for each species. A higher proportion of the *S. mansoni* data had $m/n \rightarrow 1$ which magnifies the influence of $N_{max}$ (see Fig 4). Parameter $N_{max}$ is required to define a (here uniform) prior distribution for the number of female worms per host which in turn ensures a bounded posterior distribution. Other non-uniform priors, such as the negative binomial distribution [38], are possible, but entail more prior assumptions of unknown parameters (a mean and overdispersion parameter). Irrespective of the choice of prior, it is intuitive that when $m/n \rightarrow 1$ (i.e. each offspring/miracidia in a sample is identified as coming from a unique female genotype) the data hold no information on the likely number of female worms and therefore assumptions on the prior dominate the inference. Hence, our cautious interpretation of the *S. mansoni* data highlights the importance of robust sensitivity analysis when applying our method. It also reiterates the imperative of sampling effort to obtain as many accessible offspring parasite stages (in this case, miracidia) for genotyping as possible [38].

An additional limitation related to the nature of data is that we could not account for the possibility that some individuals may have been sampled repeatedly, leading to potential correlation among these repeated measures. For the Zanzibarian (SCORE/ZEST) study, data were recorded as independent cross-sections at each sampling time. That is, at each sample collection time, data from each individual were assigned a new identifier, but this was not matched (for individuals who may have been sampled repeatedly) between different years. Nevertheless, the percentage of individuals contributing repeated measures is likely to be low, because both the participants and the subset of participants contributing samples for population genetic analysis were randomly selected at each year. For the Tanzanian (SCI) study, the percentage of individuals sampled in 2005 and followed up in 2006 was 2.5% in one school, and 19% in the other school [40]. In 2010, this percentage is likely to be even lower, since only the children who were 7 or 8 years old in 2005 or 2006 could have been resampled in 2010.

A key methodological aspect of our approach is the use of a regression calibration technique to account for inherent bias and uncertainty in the number of inferred female schistosomes.

The number of unique worm genotypes identified using sibship reconstruction is always an underestimate of the true number of reproductively active females because of the finite miracidial sample size. Similar to the influence of $N_{max}$, the degree of bias and uncertainty is dependent on the ratio $m/n$ [38]. Hence, as demonstrated here, it is critical that a calibration approach is used to adjust for these effects for each choice of prior parameters (here, just $N_{max}$ but priors with more parameters are possible [38]). Systematic bias (underestimation) alters the shape of the relationship between egg counts and worm burdens, potentially leading to erroneous inference if ignored.

In addition to the statistical limitations of the approach used here, there exist a variety of population biological and genetic assumptions not considered here explicitly that may affect the accuracy of sibship reconstruction. These include assumptions of monogamous versus polygamous mating and of Mendelian inheritance and Hardy Weinberg equilibrium of parental genotypes (see [38] for a more detailed discussion). Moreover, the number and type of microsatellite markers used in the analysis and the possible existence of clonal worms within hosts (due to the asexual reproduction of the parasite in snails; but note that cercariae mix in water before infecting hosts which likely mitigates this effect) are also limitations of the sibship reconstruction method. Notwithstanding, the statistical relationship between the estimated number of fecund female worms and the number of unique parental genotypes identified from a finite sample of (miracidial) offspring will be unaffected by the specific assumptions used for sibship reconstruction.

Ultimately, validation of the approach would require that the number of female worms per host inferred by sibship reconstruction be compared with directly observed counts. This could be achieved in humans where adult helminths can be enumerated directly (e.g. soil-transmitted helminths by chemoexpulsion) or in animals where parasites can be counted either by chemoexpulsion or dissection (e.g. in an abattoir setting or experimental system). For example, dissection has been used to count adult *S. mansoni* parasites in 37 olive baboons, where no density-dependent relationship between egg counts and (directly observed) worm burdens was found [46]. In principle, in any situation where offspring can be genotyped and adult worms counted directly, the posterior distribution of inferred female worm burden could be compared with the directly observed counts.

## The public health importance of density dependence

Notwithstanding the need for validation—and while we cannot profess to have resolved the question of density-dependent fecundity in human schistosomiasis—our method represents a promising new approach to evaluate density dependencies, processes which are essential for the regulation of parasite populations and profoundly influence their transmission dynamics and resilience to intervention [18–21]. Intuitively, as drug-based or other interventions reduce the size of the parasite populations, density dependencies are relaxed, transmission becomes more efficient, and it becomes harder to maintain progress towards elimination endpoints. Understanding density dependencies is thus important for predicting (modelling) the likely impact and effectiveness of intervention strategies. For example, when models are calibrated to pre-intervention (endemic) epidemiological data, the assumed severity of density dependence leads to different estimates of the basic reproduction number, $R_0$ [18], and different predictions on the intensity and duration of intervention efforts required to achieve control or elimination.

Knowing whether egg production is regulated in a density-dependent manner is also key to interpreting routine parasitological (egg count) monitoring and evaluation data, because for schistosomiasis—and indeed many other helminthiases—egg counts are used as a proxy for

infection intensity. The WHO goals for controlling schistosomiasis morbidity and achieving elimination as a public health problem are defined as reaching less than 1% prevalence of heavy-intensity infections in school-aged children (SAC; 5–14 years old). Countries that achieve morbidity control can progress towards interruption of transmission [10]. Thus, the thresholds used to define light and heavy intensity *S. haematobium* infections (50 eggs/10ml urine), and light, medium and heavy *S. mansoni* infections (100 and 400 epg respectively) [11] provide empirical benchmarks for decision-making on how long MDA should be maintained. Yet depending on the (non-linear) relationship between egg counts and infection intensity (worm burden), the thresholds will have different meanings for the likelihood of sustained control or resurgent infection [37, 47]. As interventions progress and interruption of transmission becomes possible in some foci [11], targets with a stronger alignment to parasite breakpoints—which integrate facilitating (parasite mating) and constraining density dependencies [20, 21, 48]—may be necessary.

Density-dependent fecundity in human helminthiases has perhaps been best studied in *Ascaris lumbricoides* [21, 49], where adult worms are easily accessible by chemoexpulsion techniques. Interestingly, although density dependence is found consistently, both the reproductive output of female worms and the severity of density dependence has been found to be highly geographically variable [49]. This poses a substantial challenge to predicting the impact of interventions in different locations and also to interpreting egg count data in terms of control and elimination prospects. Data are currently too scarce to determine whether the reproductive output of schistosomes varies geographically (although here we did find differences in the fecundity of *S. haematobium* between Pemba Island and Unguja and of *S. mansoni* between schools in Tanzania, see Tables A and B in S1 Text). However, the method outlined in this paper—combining parasitological and genotypic data—could be used to explore this and potentially provide new insights for informing egg count thresholds that are tailored to specific settings or locations.

Improving our understanding of density-dependent population processes also has broader and longer-term importance for safeguarding the effectiveness of MDA programmes, not just for schistosomiasis but other helminthiases with heavy reliance on chemotherapeutic control. Density dependencies could enhance the spread of emerging drug resistance, due to enhanced reproduction rates in resistant or reduced-susceptibility parasites [50]. In the event of emerging resistance, understanding density-dependent processes would thus be key in understanding the likely rate of spread, informing timeframes for implementing alternative or complementary interventions, such as snail control [51] or eventually a vaccine [52, 53]. Density dependencies may also complicate the interpretation of data from vaccine trials [53] and from efficacy monitoring activities [54] if interpretation is predicated on a proportional relationship between adult schistosomes and egg output (e.g. the release of fecundity constraints on surviving parasites after treatment could be falsely interpreted as reduced drug efficacy [54]).

The importance of density-dependent processes on the population biology and transmission dynamics of schistosomiasis and other helminthiases is clear and much work (including this paper) focuses on phenomenological identification, often for the purpose of integrating into mathematical models. Less clear are the biological mechanisms that drive these processes. The most straightforward explanations are putative crowding effects and competition for resources. But in schistosomiasis, more complex mechanisms may also be involved. For example, epidemiological and modelling studies have shown evidence for anti-fecundity immunity such that egg production is reduced by the host's immune response, albeit susceptibility to new infections is sustained [55, 56]. In baboons, it has been shown that adult worm pairs that had stopped egg production in a chronic infection started excreting eggs when they were

transplanted into a naïve baboon [57]. Anti-fecundity immunity has also been shown in cattle infected with *S. bovis* [58, 59]. It has also been suggested that worm fecundity declines with worm age (reproductive senescence) and that adult worms stimulate immune responses against cercarial infection (concomitant immunity) [60].

## Concluding remarks

We have revisited the longstanding and unresolved question of density-dependent fecundity in human schistosome infections using an approach that combines information from molecular and parasitological data within a robust statistical framework. We provide the first clear evidence for the operation of density-dependent fecundity in *S. haematobium*, the cause of millions of cases of urogenital schistosomiasis. In addition, we illustrate how our approach can be applied more generally to investigate density dependencies in other helminthiases where adult parasites are inaccessible. Density dependencies are key determinants of the resilience of helminthiases to interventions and therefore our findings are important both to mathematical modellers predicting the impact of intervention strategies, but also to public policy makers striving to meet the 2030 elimination targets for schistosomiasis.

## Supporting information

**S1 Text. Text A. Model variants with interactive terms permitting the severity of density dependence to vary with the other measured covariates. Fig A. Coefficient of density dependence, $\beta_1$, estimated from data on *Schistosoma mansoni* egg counts and numbers of female worms inferred by sibship reconstruction**. In panels 1 and 2 each point indicates, respectively, the point estimate and upper 95% confidence limit of the coefficient of density dependence estimated by repeatedly fitting a log-linear regression model to 1,000 datasets. Each dataset was generated by sampling from the posterior distribution of the inferred number of female worms, $N$, assuming a different value $N_{\max}$ for the upper bound of the uniform prior distribution of $N$. The box and whiskers depict the median, interquartile range and the 2.5th and 97.5th percentiles of the estimates. Values less than 1 indicate statistical support for density-dependent fecundity. **Table A. Coefficient estimates from the log-linear statistical model fitted to the *Schistosoma haematobium* data**. Coefficient estimates are arithmetic means obtained by repeatedly re-fitting the model to 1,000 datasets. Each dataset was generated by sampling from the posterior distribution of the inferred number of female worms, $N$, assuming a different value $N_{\max}$ for the upper bound of the uniform prior distribution of $N$. The coefficients $\beta_0$, $\beta_1$, $\gamma_3$, and $\gamma_4$ were statistically significant, showing that worm fecundity was higher in Pemba Island and in children, and showing evidence for density-dependent fecundity. The reference levels used for each factor variable were Pemba Island ($\gamma_2$), child ($\gamma_3$), 2012 ($\gamma_4$) and male ($\gamma_5$). The averaged ±95%CIs across all $N_{\max}$ values were $\beta_0$ (2.96, 3.94), $\beta_1$ (3.04, 4.22), $\gamma_2$ year 2016 (2.51, 4.17), $\gamma_3$ Unguja (1.37, 2.97), $\gamma_4$ adult (1.87,3.64) and $\gamma_5$ female (2.39, 3.99). **Table B. Coefficient estimates from the log-linear statistical model fitted to the *Schistosoma mansoni* data**. Coefficient estimates are arithmetic means obtained by repeatedly re-fitting the model to 1,000 datasets. Each dataset was generated by sampling from the posterior distribution of the inferred number of female worms, $N$, assuming a different value $N_{\max}$ for the upper bound of the uniform prior distribution of $N$. The coefficients $\gamma_2$ and $\gamma_3$ were statistically significant, showing that worm fecundity was higher in 2006 and 2010, and Kisorya school. The reference levels used for each factor variable were 2005 ($\gamma_2$), Bukindo ($\gamma_3$) and male ($\gamma_4$). The averaged ±95%CIs across all $N_{max}$ values were $\beta_0$ (0.01,3.62), $\beta_1$ (-0.10,4.18), $\gamma_2$ year 2016 (0.41, 5.65), $\gamma_2$ year 2010 (1.58, 7.33), $\gamma_3$ Kisorya (0.95, 5.97), $\gamma_4$ female (-0.99, 4.0). **Table C. Results of likelihood ratio tests comparing the fits of models with additive or**

**interactive terms**. Likelihood ratio tests (LRTs) were used to compare the fits of models including either additive or interactive terms that were re-fitted to 1,000 datasets. Each dataset was generated by sampling from the posterior distribution of the inferred number of female worms, $N$, assuming a different value $N_{max}$ for the upper bound of the uniform prior distribution of $N$. The frequency with which fitted models including interactive terms were preferred over the simpler model including only additive terms is given in the column labelled "Percentage preference for interactive model". Results are shown for *Schistosoma haematobium* in Zanzibar in panel 1 and for *S. mansoni* in mainland Tanzania in panel 2.
(DOCX)

**S1 Data. Tanzanian data used for this study [31].**
(CSV)

## Acknowledgments

We thank Dr Stefanie Knopp, Professor David Rollinson and Dr Tom Pennance for sharing and/or assistance preparing the Zanzibar dataset with us and for their feedback on the manuscript.

## Author Contributions

**Conceptualization:** M. Inês Neves, Martin Walker.

**Data curation:** Charlotte M. Gower, Joanne P. Webster.

**Formal analysis:** M. Inês Neves.

**Funding acquisition:** M. Inês Neves, Charlotte M. Gower, Joanne P. Webster, Martin Walker.

**Investigation:** M. Inês Neves, Charlotte M. Gower, Joanne P. Webster, Martin Walker.

**Methodology:** M. Inês Neves, Martin Walker.

**Project administration:** Joanne P. Webster, Martin Walker.

**Resources:** Charlotte M. Gower, Joanne P. Webster.

**Software:** M. Inês Neves, Martin Walker.

**Supervision:** Joanne P. Webster, Martin Walker.

**Validation:** M. Inês Neves, Martin Walker.

**Visualization:** M. Inês Neves.

**Writing – original draft:** M. Inês Neves, Martin Walker.

**Writing – review & editing:** M. Inês Neves, Charlotte M. Gower, Joanne P. Webster, Martin Walker.

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
