## [Decision Letter · Decision Letter 0]

22 Feb 2021

Dear Miss Neves,

Thank you very much for submitting your manuscript "Revisiting density-dependent fecundity in schistosomes using sibship reconstruction" for consideration at PLOS Neglected Tropical Diseases. As with all papers reviewed by the journal, your manuscript was reviewed by members of the editorial board and by several independent reviewers. The reviewers appreciated the attention to an important topic. Based on the reviews, we are likely to accept this manuscript for publication, providing that you modify the manuscript according to the review recommendations. 

Sincerely,

Brianna R Beechler, Ph.D., DVM

Associate Editor

Timothy Geary

Deputy Editor

Reviewer's Responses to Questions

**Key Review Criteria Required for Acceptance?**

**Methods**

-Are the objectives of the study clearly articulated with a clear testable hypothesis stated?

-Is the study design appropriate to address the stated objectives?

-Is the population clearly described and appropriate for the hypothesis being tested?

-Is the sample size sufficient to ensure adequate power to address the hypothesis being tested?

-Were correct statistical analysis used to support conclusions?

-Are there concerns about ethical or regulatory requirements being met?

Reviewer #1: The study design addresses the stated objectives.

Two existing datasets, ZEST (Zanzibar) and SCORE (Tanzania) were used for S. haematobium and S. mansoni, respectively.

Sample sizes are sufficient: ZEST = 224 children and adults before treatment and 214 children and adults after 10 rounds of MDA. SCORE = 151 school children before and during MDA treatment. My biggest concern is whether the analysis included repeated measures of the same individual and how the authors accounted for that, especially samples collected after treatment (MDA)?

The miracidia used to determine sibship reconstruction had low sample sizes. However, I assume that it is the nature of the sampling schemes for both schistosoma species. In other words, normal numbers expected with egg hatching detection methods from urine and stool samples. The authors did not include details on the egg hatching technique. Are there differences between egg hatching from urine samples or stool samples? What was the egg hatching success rate? 

Other than my concern mentioned above, the statistical approaches seemed applicable and limitations and assumptions were clearly stated. 

There are no IRB approved references – however the use of human subjects in this study do not necessarily require and IRB approval. Although, institutional differences might exist.

Reviewer #2: Overall the methodology is clearly described. However, the Methods should contain a brief description of how the sibship reconstruction is achieved, without having to refer to the relevant references. This is not as universally understood as immunoblotting or PCR. This would also help illuminate the subsequent discussion of the limitations of sibship reconstruction presented in the Discussion.

Reviewer #3: (No Response)

**Results**

-Does the analysis presented match the analysis plan?

-Are the results clearly and completely presented?

-Are the figures (Tables, Images) of sufficient quality for clarity?

Reviewer #1: The simpler additive models included indicator variables that was not reported on in great detail (I do understand that it was not the main focus of the paper). However, I suggested a supplementary text box extrapolating on all the significant indicator variables that were presented in Tables S1 and S2. Age differences, location differences, etc. How the indicator variables are influenced by the observed eggs (epg or eggs per 10ml urine)? Etc. 

The authors also noted that the models including interaction terms were less parsimonious. So why did the authors decide to use the simpler additive models? Perhaps they could provide a clear explanation supporting their decision.

The figures and tables seem to be of sufficient quality.

Reviewer #2: The Results are clearly and concisely described. No concerns.

Reviewer #3: (No Response)

**Conclusions**

-Are the conclusions supported by the data presented?

-Are the limitations of analysis clearly described?

-Do the authors discuss how these data can be helpful to advance our understanding of the topic under study?

-Is public health relevance addressed?

Reviewer #1: The authors conclusions aligned with the data presented and limitations of the models were clearly stated.

Reviewer #2: The Conclusions are clearly and succinctly communicated, without over-interpretation. I especially appreciated the thorough and objective discussion of the study’s limitations. The potential relevance of the findings to model refinement as well as real-world efforts to control schistosomiasis are fascinating and the authors are correct to highlight these points.

Reviewer #3: (No Response)

**Editorial and Data Presentation Modifications?**

Reviewer #1: Major issues:

How did the authors accounted for repeated measures of individual samples used (before and after treatment)?

Minor issues:

Minor comments/edits – see specific lines under headings.

Suggesting a Supplementary text box extrapolating on all the significant indicator variables that were presented in Tables S1 and S2. Age differences, location differences, etc. How the indicator variables are influenced by the observed eggs (epg or eggs per 10ml urine)?

Abstract:

N/A

Author summary:

Line 52: Perhaps the authors could be more descriptive to the “they” they’re referring to. Should it be “We”?

Background:

N/A

Materials and Methods:

Line 145: I wonder if it is possible to briefly explain the specific details on the molecular analysis and sibship reconstruction?

Line 146 Statistical modeling: General comment for this section: How did the authors account for repeated measures of the same individual (data collection from before and after treatment as explained under lines 121-126)? Individual ID must be included as a random effect if repeated measures of the same individual exist.

Results:

Line 180-188: I would suggest this whole section move to methods rather than results. Although, rephrasing and word choice could also improve the fit as results. 

Line 186: Start the sentence “We used a…” with Briefly, we used a… Assuming here that the authors explaining the “recently developed statistical approach”.

Line 217-219: Are there any interesting results that could be reported here from the supplementary outputs? Also, see my minor issues suggestion.

Line 221-222: If the interaction term models were less parsimonious why did the authors decide to use the simpler additive models?

Discussion:

Line 227: Rephrase (e.g., We found supportive evidence that clearly demonstrated density-dependent fecundity in S. haematobium, the causative agent responsible for urogenital schistosomiasis).

Line 232: Rephrase (e.g., “… and can be extended to other human helminthiases…”

Line 244: Rephrase (e.g., “…A recent study by Gowler et al. [37], utilizing the same Tanzaniam S. mansoni dataset, found …”).

Line 255-262: I would suggest re-organizing content in the paragraph(s). Perhaps starting as follow: Contradictory results exist for density-dependent fecundity in S. haematobium. One study analyzed 197 infected cases at autopsy found no significant evidence of density dependence [27]. Whereas a relative recent study… dependence [43].

Line 261: Move up to previous paragraph (line 260).

Line 262: Start sentence with “Firstly, in…”

Line 264: Start sentence with “Secondly, it is possible…”

Line 265: Delete “On the other hand” Perhaps phrase as follow: Although our non-invasive method of exploring density dependence is more indirect (but practicable), it does invoke modeling assumptions (to which… limitation.

Line 282: What do the authors mean by “obtaining as many accessible parasite stages”? 

Line 306: Perhaps the authors could reiterate the “specific assumptions” here. 

Line 313: S. mansoni should be in italics

Line 320: Delete “we believe that”

Line 321: Replace evaluating with evaluate

Line 355: In what context does the authors use “the same egg counts measured”? Do the authors infer that the same egg count method is used or is it the variation having different individuals doing the egg counts?

Line 359: Add “to” (e.g., …be used to explore…)

Line 366: Delete “that”

Line 394: Suggestion: “…schistosomiasis. In addition, we illustrate how our approach…”

Figures:

Figure 3: Out of curiosity, what was the criteria for the Nmax limit of 250.

Tables:

Table S1 and Tables S2: I would suggest adding the reference levels for the indicator variables to your table caption. 

Table S3: Caption reads strangely – Perhaps: “… (LRT) at different values of Nmax for A) S. haematobium in Zanzibar and B) S. mansoni in mainland Tanzania.”

Reviewer #2: Various minor typographical issues:

Line 66 – “Globally, schistosomiasis is second only to malaria in terms of socioeconomic impact [5].” Among eukaryotic pathogens, that is.

Line 76 – “if necessary”

Line 85 – “utmost importance”

Line 227 – “which is the first time”

Line 229 – “the cause of intestinal schistosomiasis”

Line 359 – “used to explore”

Line 365 – “In the event that of emerging”?

Line 367 – “rate of spread, informing”

Line 381 – “immune response, albeit”

Reviewer #3: (No Response)

**Summary and General Comments**

Reviewer #1: Overall, I found this manuscript to be well written. I find the topic interesting and it contributes substantially to the literature, especially the non-invasive nature of the methods. I have mainly minor issues, but one major concern I have is whether the two datasets included repeated measures from individual subjects. The methods mentioned before and after treatment collections of individuals. How did the authors accounted for samples who were treated, as treatment could influence density dependence.

Reviewer #2: Overall, this is a well-written and highly readable manuscript which provides convincing evidence for the existence of density-dependent fecundity in Schistosoma haematobium infections. Combining molecular genotyping and traditional parasitological data with sophisticated modeling, this study is a logical extension of the authors’ previous work and should be of considerable interest to the schistosomiasis community, and to helminthologists in general.

Reviewer #3: This paper investigates an important matter in schistosome infections, namely, does the number of adult worms affect fecundity of individual females. This is an important question, because of the intense pathogenesis of eggs laid by the females. The work is built around some elegant genetic typing of miracidia combined with statistical analyses.

In essence, the paper presents a useful case and is worthy of consideration after some issues are addressed.

Could the authors comment on:

1. The proximity of the analyses to drug administration in community treatments. The surveys were performed in sites that had been treated regularly with praziquantel. What is the effect of PZQ on population structure in communities and in individuals? What would be the effect of, say, killing of some, but not all, worms in humans by PZQ?

2. Is the immune or health status of the human hosts likely to confound results? 

3. Recently, there have been some imaging and immunological methods suggested to estimate worm burdens in the host. perhaps some reference to these could be made, if only to comment on their usefulness.

PLOS authors have the option to publish the peer review history of their article (what does this mean?). If published, this will include your full peer review and any attached files.

Reviewer #1: No

Reviewer #2: Yes: Stephen J. Davies

Reviewer #3: No

Figure Files:

Data Requirements:

Reproducibility:

References

---

## [Decision Letter · Decision Letter 1]

19 Apr 2021

Dear Miss Neves,

We are pleased to inform you that your manuscript 'Revisiting density-dependent fecundity in schistosomes using sibship reconstruction' has been provisionally accepted for publication in PLOS Neglected Tropical Diseases.

Best regards,

Brianna R Beechler, Ph.D., DVM

Associate Editor

Timothy Geary

Deputy Editor

Reviewer one noted a few typographical errors that should be fixed prior to publication.

Reviewer's Responses to Questions

**Key Review Criteria Required for Acceptance?**

**Methods**

-Are the objectives of the study clearly articulated with a clear testable hypothesis stated?

-Is the study design appropriate to address the stated objectives?

-Is the population clearly described and appropriate for the hypothesis being tested?

-Is the sample size sufficient to ensure adequate power to address the hypothesis being tested?

-Were correct statistical analysis used to support conclusions?

-Are there concerns about ethical or regulatory requirements being met?

Reviewer #1: (No Response)

Reviewer #2: (No Response)

Reviewer #3: (No Response)

**Results**

-Does the analysis presented match the analysis plan?

-Are the results clearly and completely presented?

-Are the figures (Tables, Images) of sufficient quality for clarity?

Reviewer #1: (No Response)

Reviewer #2: (No Response)

Reviewer #3: (No Response)

**Conclusions**

-Are the conclusions supported by the data presented?

-Are the limitations of analysis clearly described?

-Do the authors discuss how these data can be helpful to advance our understanding of the topic under study?

-Is public health relevance addressed?

Reviewer #1: (No Response)

Reviewer #2: (No Response)

Reviewer #3: (No Response)

**Editorial and Data Presentation Modifications?**

Reviewer #1: The authors have addressed all of my concerns I had and I accept their explanations and revised manuscript.

Typos on lines:

Line 132: ...urine.. should be urine.

Line 669: 0,01 should be 0.01

Line 680: both areas refer to panel B - one should be panel A

Reviewer #2: (No Response)

Reviewer #3: (No Response)

**Summary and General Comments**

Reviewer #1: (No Response)

Reviewer #2: (No Response)

Reviewer #3: (No Response)

PLOS authors have the option to publish the peer review history of their article (what does this mean?). If published, this will include your full peer review and any attached files.

Reviewer #1: No

Reviewer #2: No

Reviewer #3: No

---

## [Editor Report · Acceptance letter]

10 May 2021

Dear Miss Neves,

We are delighted to inform you that your manuscript, "Revisiting density-dependent fecundity in schistosomes using sibship reconstruction," has been formally accepted for publication in PLOS Neglected Tropical Diseases.

Best regards,

Shaden Kamhawi

co-Editor-in-Chief

Paul Brindley

co-Editor-in-Chief
